# Beyond the Birkhoff Polytope:
# Convex Relaxations for Vector Permutation Problems

**Cong Han Lim**
Department of Computer Sciences
University of Wisconsin - Madison
Madison, WI 53706
conghan@cs.wisc.edu

**Stephen J. Wright**
Department of Computer Sciences
University of Wisconsin - Madison
Madison, WI 53706
swright@cs.wisc.edu

## Abstract

The Birkhoff polytope (the convex hull of the set of permutation matrices), which is represented using $\Theta(n^2)$ variables and constraints, is frequently invoked in formulating relaxations of optimization problems over permutations. Using a recent construction of Goemans [1], we show that when optimizing over the convex hull of the permutation vectors (the permutahedron), we can reduce the number of variables and constraints to $\Theta(n \log n)$ in theory and $\Theta(n \log^2 n)$ in practice. We modify the recent convex formulation of the 2-SUM problem introduced by Fogel et al. [2] to use this polytope, and demonstrate how we can attain results of similar quality in significantly less computational time for large $n$. To our knowledge, this is the first usage of Goemans' compact formulation of the permutahedron in a convex optimization problem. We also introduce a simpler regularization scheme for this convex formulation of the 2-SUM problem that yields good empirical results.

## 1 Introduction

A typical workflow for converting a discrete optimization problem over the set of permutations of $n$ objects into a continuous relaxation is as follows: (1) use permutation matrices to represent permutations; (2) relax to the convex hull of the set of permutation matrices — the Birkhoff polytope; (3) relax other constraints to ensure convexity/continuity. Instances of this procedure appear in [3, 2]. Representation of the Birkhoff polytope requires $\Theta(n^2)$ variables, significantly more than the $n$ variables required to represent the permutation directly. The increase in dimension is unappealing, especially if we are only interested in optimizing over permutation vectors, as opposed to permutations of a more complex object, such as a graph. The obvious alternative of using a relaxation based on the convex hull of the set of permutations (the permutahedron) is computationally infeasible, because the permutahedron has exponentially many facets (whereas the Birkhoff polytope has only $n^2$ facets). We can achieve a better trade-off between the number of variables and facets by using sorting networks to construct polytopes that can be linearly projected to recover the permutahedron. This construction, introduced by Goemans [1], can have as few as $\Theta(n \log n)$ facets, which is optimal up to constant factors. In this paper, we use a relaxation based on these polytopes, which we call "sorting network polytopes."

We apply the sorting network polytope to the *noisy seriation problem*, defined as follows. Given a noisy similarity matrix $A$, recover a symmetric row/column ordering of $A$ for which the entries generally decrease with distance from the diagonal. Fogel et al. [2] introduced a convex relaxation of the 2-SUM problem to solve the noisy seriation problem. They proved that the solution to the 2-SUM problem recovers the exact solution of the seriation problem in the "noiseless" case (in which an ordering exists that ensures monotonic decrease of similarity measures with distance from the diagonal). They further show that the formulation allows side information about the ordering to be incorporated, and is more robust to noise than a spectral formulation of the 2-SUM problem de-

scribed by Atkins et al. [4]. The formulation in [2] makes use of the Birkhoff polytope. We propose instead a formulation based on the sorting network polytope. Performing convex optimization over the sorting network polytope requires different techniques from those described in [2]. In addition, we describe a new regularization scheme, applicable both to our formulation and that of [2], that is more natural for the 2-SUM problem and has good practical performance.

The paper is organized as follows. We begin by describing polytopes for representing permutations in Section 2. In Section 3, we introduce the seriation problem and the 2-SUM problem, describe two continuous relaxations for the latter, (one of which uses the sorting network polytope) and introduce our regularization scheme for strengthening the relaxations. Issues that arise in using the sorting network polytope are discussed in Section 4. In Section 5, we provide experimental results showing the effectiveness of our approach. The extended version of this paper [5] includes some additional computational results, along with several proofs. It also describes an efficient algorithm for taking a conditional gradient step for the convex formulation, for the case in which the formulation contains no side information.

## 2   Permutahedron, Birkhoff Polytope, and Sorting Networks

We use $n$ throughout the paper to refer to the length of the permutation vectors. $\pi_{I_n} = (1, 2, \ldots, n)^T$ denotes the identity permutation. (When the size $n$ can be inferred from the context, we write the identity permutation as $\pi_I$.) $\mathcal{P}^n$ denotes the set of all permutations vectors of length $n$. We use $\pi \in \mathcal{P}^n$ to denote a generic permutation, and denote its components by $\pi(i)$, $i = 1, 2, \ldots, n$. We use $\mathbf{1}$ to denote the vector of length $n$ whose components are all 1.

**Definition 2.1.** *The* permutahedron $\mathcal{PH}^n$, *the convex hull of* $\mathcal{P}^n$, *is defined as follows:*

$$\mathcal{PH}^n := \left\{ x \in \mathbb{R}^n \;\middle|\; \sum_{i=1}^{n} x_i = \frac{n(n+1)}{2}, \sum_{i \in S} x_i \leq \sum_{i=1}^{|S|} (n+1-i) \text{ for all } S \subset [n] \right\}.$$

The permutahedron $\mathcal{PH}^n$ has $2^n - 2$ facets, which prevents us from using it in optimization problems directly. (We should note however that the permutahedron is a submodular polyhedron and hence admits efficient algorithms for certain optimization problems.) Relaxations are commonly derived from the set of permutation matrices (the set of $n \times n$ matrices containing zeros and ones, with a single one in each row and column) and its convex hull instead.

**Definition 2.2.** *The convex hull of the set of* $n \times n$ *permutation matrices is the* Birkhoff polytope $\mathcal{B}^n$, *which is the set of all doubly-stochastic* $n \times n$ *matrices* $\{X \in \mathbb{R}^{n \times n} \mid X \geq 0, X\mathbf{1} = \mathbf{1}, X^T\mathbf{1} = \mathbf{1}\}$.

The Birkhoff polytope has been widely used in the machine learning and computer vision communities for various permutation problems (see for example [2], [3]). The permutahedron can be represented as the projection of the Birkhoff polytope from $\mathbb{R}^{n \times n}$ to $\mathbb{R}^n$ by $x_i = \sum_{j=1}^{n} j \cdot X_{ij}$. The Birkhoff polytope is sometimes said to be an *extended formulation* of the permutahedron.

A natural question to ask is whether a more compact extended formulation exists for the permutahedron. Goemans [1] answered this question in the affirmative by constructing one with $\Theta(n \log n)$ constraints and variables, which is optimal up to constant factors. His construction is based on *sorting networks*, a collection of wires and binary comparators that sorts a list of numbers. Figure 1 displays a sorting network on 4 variables. (See [6] for further information on sorting networks.)

Given a sorting network on $n$ inputs with $m$ comparators (we will subsequently always use $m$ to refer to the number of comparators), an extended formulation for the permutahedron with $O(m)$ variables and constraints can be constructed as follows [1]. Referring to the notation in the right subfigure in Figure 1, we introduce a set of constraints for each comparator $k = 1, 2, \ldots, m$ to indicate the relationships between the two inputs and the two outputs of each comparator:

$$x^k_{(\text{in, top})} + x^k_{(\text{in, bot})} = x^k_{(\text{out, top})} + x^k_{(\text{out, bot})}, \quad x^k_{(\text{out, top})} \leq x^k_{(\text{in, top})}, \quad \text{and} \quad x^k_{(\text{out, top})} \leq x^k_{(\text{in, bot})}. \quad (1)$$

Note that these constraints require the sum of the two inputs to be the same as the sum of the two outputs, but the inputs can be closer together than the outputs. Let $x^{\text{in}}_i$ and $x^{\text{out}}_i$, $i = 1, 2, \ldots, n$ denote the $x$ variables corresponding to the $i$th input and $i$th output of the entire sorting network, respectively. We introduce the additional constraints

$$x^{\text{out}}_i = i, \text{ for } i \in [n]. \quad (2)$$

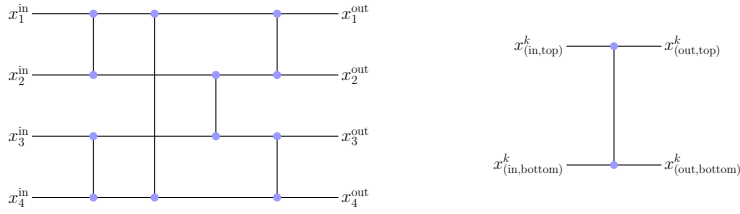

Figure 1: A bitonic sorting network on 4 variables (left) and the $k$-th comparator (right). The input to the sorting network is on the left and the output is on the right. At each comparator, we take the two input values and sort them such that the smaller value is the one at the top in the output. Sorting takes place progressively as we move from left to right through the network, sorting pairs of values as we encounter comparators.

The details of this construction depend on the particular choice of sorting network (see Section 4), but we will refer to it generically as the *sorting network polytope* $\mathcal{SN}^n$. Each element in this polytope can be viewed as a concatenation of two vectors: the subvector associated with the network inputs $x^{\text{in}} = (x_1^{\text{in}}, x_2^{\text{in}}, \ldots, x_n^{\text{in}})$, and the rest of the coordinates $x^{\text{rest}}$, which includes all the internal variables as well as the outputs. The following theorem attests to the fact that any input vector $x^{\text{in}}$ vector that is part of a feasible vector for the entire network is a point in the permutahedron:

**Theorem 2.3** (Goemans [1])**.** *The set $\left\{ x^{in} \mid (x^{in}, x^{rest}) \in \mathcal{SN}^n \right\}$ is the permutahedron $\mathcal{PH}^n$.*

## 3    Convex Relaxations of 2-SUM via Sorting Network Polytope

In this section we will briefly describe the seriation problem, and some of the continuous relaxations of the combinatorial 2-SUM problem that can be used to solve this problem.

**The Noiseless Seriation Problem.**    The term *seriation* generally refers to data analysis techniques that arrange objects in a linear ordering in a way that fits available information and thus reveals underlying structure of the system [7]. We adopt here the definition of the *seriation problem* from [4]. Suppose we have $n$ objects arranged along a line, and a similarity function that increases with distance between objects in the line. The similarity matrix is the symmetric $n \times n$ matrix whose $(i, j)$ entry is the similarity measure between the $i$th and $j$th objects in the linear arrangement. This similarity matrix is a *R-matrix*, according to the following definition.

**Definition 3.1.** *A symmetric matrix $A$ is a* Robinson matrix *(R-matrix) if for all points $(i, j)$ where $i > j$, we have $A_{ij} \leq \min(A_{(i-1),j}, A_{i,(j+1)})$. A symmetric matrix $A$ is a* pre-R *matrix if $\Pi^T A \Pi$ is R for some permutation $\Pi$.*

In other words, a symmetric matrix is a R-matrix if the entries are nonincreasing as we move away from the diagonal in either the horizontal or vertical direction. The goal of the *noiseless seriation problem* is to recover the ordering of the variables along the line from the pairwise similarity data, which is equivalent to finding the permutation that recovers an R-matrix from a pre-R-matrix.

The seriation problem was introduced in the archaeology literature [8], and has applications across a wide range of areas including clustering [9], shotgun DNA sequencing [2], and taxonomy [10]. R-matrices are useful in part because of their relation to the *consecutive-ones property* in a matrix of zeros and ones, where the ones in each column form a contiguous block. A matrix $M$ with the consecutive-ones property gives rise to a R-matrix $MM^T$.

**Noisy Seriation, 2-SUM and Continuous Relaxations.**    Given a binary symmetric matrix $A$, the 2-SUM problem can be expressed as follows:

$$\min_{\pi \in \mathcal{P}^n} \ \sum_{i=1}^{n} \sum_{j=1}^{n} A_{ij} (\pi(i) - \pi(j))^2. \tag{3}$$

A slightly simpler but equivalent formulation, defined via the Laplacian $L_A = \text{diag}(A\mathbf{1}) - A$, is

$$\min_{\pi \in \mathcal{P}^n} \ \pi^T L_A \pi. \tag{4}$$

The seriation problem is closely related to the combinatorial 2-SUM problem, and Fogel et al. [2] proved that if $A$ is a pre-$R$-matrix such that each row/column has unique entries, then the solution to the 2-SUM problem also solves the noiseless seriation problem. In another relaxation of the 2-SUM problem, Atkins et al. [4] demonstrate that finding the second smallest eigenvalue, also known as the *Fiedler value*, solves the noiseless seriation problem. Hence, the 2-SUM problem provides a good model for the *noisy* seriation problem, where the similarity matrices are close to, but not exactly, pre-R matrices.

The 2-SUM problem is known to be $NP$-hard [11], so we seek efficient relaxations. We describe below two continuous relaxations that are computationally practical. (Other relaxations of these problems require solution of semidefinite programs and are intractable in practice for large $n$.)

The spectral formulation of [4] seeks the Fiedler value by searching over the space orthogonal to the vector $\mathbf{1}$, which is the eigenvector that corresponds to the zero eigenvalue. The Fiedler value is the optimal objective value of the following problem:

$$\min_{y \in \mathbb{R}^n} \ y^T L_A y \quad \text{such that} \quad y^T \mathbf{1} = 0, \ \ \|y\|_2 = 1. \tag{5}$$

This problem is non-convex, but its solution can be found efficiently from an eigenvalue decomposition of $L_A$. With Fiedler vector $y$, one can obtain a candidate solution to the 2-SUM problem by picking the permutation $\pi \in \mathcal{P}^n$ to have the same ordering as the elements of $y$. The spectral formulation (5) is a continuous relaxation of the 2-SUM problem (4).

The second relaxation of (4), described by Fogel et al. [2], makes use of the Birkhoff polytope $\mathcal{B}^n$. The basic version of the formulation is

$$\min_{\Pi \in \mathcal{B}^n} \ \pi_I^T \Pi^T L_A \Pi \pi_I, \tag{6}$$

(recall that $\pi_I$ is the identity permutation $(1, 2, \ldots, n)^T$), which is a convex quadratic program over the $n^2$ components of $\Pi$. Fogel et al. augment and enhance this formulation as follows.

- Introduce a "tiebreaking" constraint $e_1^T \Pi \pi_I + 1 \leq e_n^T \Pi \pi_I$ to resolve ambiguity about the direction of the ordering, where $e_k = (0, \ldots, 0, 1, 0, \ldots, 0)^T$ with the 1 in position $k$.
- Average over several perturbations of $\pi_I$ to improve robustness of the solution.
- Add a penalty to maximize the Frobenius norm of the matrix $\Pi$, which pushes the solution closer to a vertex of the Birkhoff polytope.
- Incorporate additional ordering constraints of the form $x_i - x_j \leq \delta_k$, to exploit prior knowledge about the ordering.

With these modifications, the problem to be solved is

$$\min_{\Pi \in \mathcal{B}^n} \ \frac{1}{p} \text{Trace}(Y^T \Pi^T L_A \Pi Y) - \frac{\mu}{p} \|P\Pi\|_F^2 \quad \text{such that} \quad D\Pi \pi_I \leq \delta, \tag{7}$$

where each column of $Y \in \mathbb{R}^{n \times p}$ is a slightly perturbed version of a permutation,[1] $\mu$ is the regularization coefficient, the constraint $D\Pi \pi_I \leq \delta$ contains the ordering information and tiebreaking constraints, and the operator $P = I - \frac{1}{n} \mathbf{1}\mathbf{1}^T$ is the projection of $\Pi$ onto elements orthogonal to the all-ones matrix. The penalization is applied to $\|P\Pi\|_F^2$ rather than to $\|\Pi\|_F^2$ directly, thus ensuring that the program remains convex if the regularization factor is sufficiently small (for which a sufficient condition is $\mu < \lambda_2(L_A)\lambda_1(YY^T)$). We will refer to this regularization scheme as the *matrix-based regularization*, and to the formulation (7) as the *matrix-regularized Birkhoff-based convex formulation*.

Figure 2 illustrates the permutahedron in the case of $n = 3$, and compares minimization of the objective $y^T L_A y$ over the permutahedron (as attempted by the convex formulation) with minimization of the same objective over the constraints in the spectral formulation (5). The spectral method returns good solutions when the noise is low, and it is computationally efficient since there are many fast algorithms and software for obtaining selected eigenvectors. However, the Birkhoff-based convex formulation can return a solution that is significantly better in situations with high noise or significant additional ordering information. For the rest of this section, we will focus on the convex formulation.

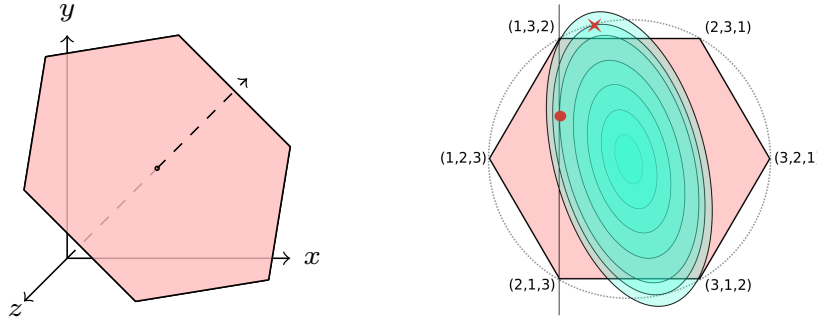

Figure 2: A geometric interpretation of spectral and convex formulation solutions on the 3-permutahedron. The left image shows the 3-permutahedron in 3D space and the dashed line shows the eigenvector $\mathbf{1}$ corresponding to the zero eigenvalue. The right image shows the projection of the 3-permutahedron along the trivial eigenvector together with the elliptical level curves of the objective function $y^T L_A y$. Points on the circumscribed circle have an $\ell_2$-norm equal to that of a permutation, and the objective is minimized over this circle at the point denoted by a cross. The vertical line in the right figure enforces the tiebreaking constraint that 1 must appear before 3 in the ordering; the red dot indicates the minimizer of the objective over the resulting triangular feasible region. Without the tiebreaking constraint, the minimizer is at the center of the permutahedron.

**A Compact Convex Relaxation via the Permutahedron/Sorting Network Polytope and a New Regularization Scheme.** We consider now a different relaxation for the 2-SUM problem (4). Taking the convex hull of $\mathcal{P}^n$ directly, we obtain

$$\min_{x \in \mathcal{PH}^n} x^T L_A x. \tag{8}$$

This is essentially a permutahedron-based version of (6). In fact, two problems are equivalent, except that formulation (8) is more compact when we enforce $x \in \mathcal{PH}$ via the sorting network constraints

$$x \in \{x^{\text{in}} \mid (x^{\text{in}}, x^{\text{rest}}) \in \mathcal{SN}^n\},$$

where $\mathcal{SN}^n$ incorporates the comparator constraints (1) and the output constraints (2). This formulation can be enhanced and augmented in a similar fashion to (6). The tiebreaking constraint for this formulation can be expressed simply as $x_1 + 1 \le x_n$, since $x^{\text{in}}$ consists of the subvector $(x_1, x_2, \ldots, x_n)$. (In both (8) and (6), having at least one additional constraint is necessary to remove the trivial solution given by the center of the permutahedron or Birkhoff polytope; see Figure 2.) This constraint is the strongest inequality that will not eliminate any permutation (assuming that a permutation and its reverse are equivalent); we include a proof of this fact in [5].

It is also helpful to introduce a penalty to force the solution $x$ to be closer to a permutation, that is, a vertex of the permutahedron. To this end, we introduce a *vector-based regularization scheme*. The following statement is an immediate consequence of strict convexity of norms.

**Proposition 3.2.** *Let $v \in \mathbb{R}^n$, and let $X$ be the convex hull of all permutations of $v$. Then, the points in $X$ with the highest $\ell_p$ norm, for $1 < p < \infty$, are precisely the permutations of $v$.*

It follows that adding a penalty to encourage $\|x\|_2$ to be large might improve solution quality. However, directly penalizing the negative of the 2-norm of $x$ would destroy convexity, since $L_A$ has a zero eigenvalue. Instead we penalize $Px$, where $P = I - \frac{1}{n}\mathbf{1}\mathbf{1}^T$ projects onto the subspace orthogonal to the trivial eigenvector $\mathbf{1}$. (Note that this projection of the permutahedron still satisfies the assumptions of Proposition 3.2.) When we include a penalty on $\|Px\|_2^2$ in the formulation (8) along with side constraints $Dx \le \delta$ on the ordering, we obtain the objective $x^T L_A x - \mu\|Px\|_2^2$ which leads to

$$\min_{x \in \mathcal{PH}^n} x^T (L_A - \mu P)x \quad \text{such that} \quad Dx \le \delta. \tag{9}$$

This objective is convex when $\mu \le \lambda_2(L_A)$, a looser condition on $\mu$ than is the case in matrix-based regularization. We will refer to (9) as the *regularized permutahedron-based convex formulation*.

Vector-based regularization can also be incorporated into the Birkhoff-based convex formulation. Instead of maximizing the $\|P\Pi\|_2^2$ term in formulation (7) to force the solution to be closer to a permutation, we could maximize $\|\tilde{P}\Pi Y\|_2^2$. The vector-regularized version of (6) with side constraints can be written as follows:

$$\min_{\Pi \in \mathcal{B}^n} \frac{1}{p} \text{Trace}(Y^T \Pi^T (L_A - \mu P) \Pi Y) \quad \text{such that} \quad D\Pi\pi_I \leq \delta. \tag{10}$$

We refer to this formulation as the *vector-regularized Birkhoff-based convex formulation*. Vector-based regularization is in some sense more natural than the regularization in (7). It acts directly on the set that we are optimizing over, rather than an expanded set. The looser condition $\mu \leq \lambda_2(L_A)$ allows for stronger regularization. Experiments reported in [5] show that the vector-based regularization produces permutations that are consistently better those obtained from the Birkhoff-based regularization.

The regularized permutahedron-based convex formulation is a convex QP with $O(m)$ variables and constraints, where $m$ is the number of comparators in its sorting network, while the Birkhoff-based one is a convex QP with $O(n^2)$ variables. The one feature in the Birkhoff-based formulations that the permutahedron-based formulations do not have is the ability to average the solution over multiple vectors by choosing $p > 1$ columns in the matrix $Y \in \mathbb{R}^{n \times p}$. However, our experiments suggested that the best solutions were obtained for $p = 1$, so this consideration was not important in practice.

## 4   Key Implementation Issues

**Choice of Sorting Network.**   There are numerous possible choices of the sorting network, from which the constraints in formulation (9) are derived. The asymptotically most compact option is the AKS sorting network, which contains $\Theta(n \log n)$ comparators. This network was introduced in [12] and subsequently improved by others, but is impractical because of its difficulty of construction and the large constant factor in the complexity expression. We opt instead for more elegant networks with slightly worse asymptotic complexity. Batcher [13] introduced two sorting networks with $\Theta(n \log^2 n)$ size — the odd-even sorting network and the bitonic sorting network — that are popular and practical. The sorting network polytope based on these can be generated with a simple recursive algorithm in $\Theta(n \log^2 n)$ time.

**Obtaining Permutations from a Point in the Permutahedron.**   Solution of the permutation-based relaxation yields a point $x$ in the permutahedron, but we need techniques to convert this point into a valid permutation, which is a candidate solution for the 2-SUM problem (3). The most obvious recovery technique is to choose this permutation $\pi$ to have the same ordering as the elements of $x$, that is, $x_i < x_j$ implies $\pi(i) < \pi(j)$, for all $i, j \in \{1, 2, \ldots, n\}$. We could also sample multiple permutations, by applying Gaussian noise to the components of $x$ prior to taking the ordering to produce $\pi$. (We used i.i.d. noise with variance $0.5$.) The 2-SUM objective (3) can be evaluated for each permutation so obtained, with the best one being reported as the overall solution. This inexpensive randomized recovery procedure can be repeated many times, and it yield significantly better results over the single "obvious" ordering.

**Solving the Convex Formulation.**   On our test machine using the Gurobi interior point solver, we were able to solve instances of the permutahedron-based convex formulation (9) of size up to around $n = 10000$. As in [2], first-order methods can be employed when the scale is larger. In [5], we provide an optimal $O(n \log n)$ algorithm for step (1), in the case in which only the tiebreaking constraint is present, with no additional ordering constraints.

## 5   Experiments

We compare the run time and solution quality of algorithms on the two classes of convex formulations — Birkhoff-based and permutahedron-based — with various parameters. Summary results are presented in this section. Additional results, including more extensive experiments comparing the effects of different parameters on the solution quality, appear in [5].

**Experimental Setup.** The experiments were run on an Intel Xeon X5650 (24 cores @ 2.66Ghz) server with 128GB of RAM in MATLAB 7.13, CVX 2.0 ([14],[15]), and Gurobi 5.5 [16]. We tested four formulation-algorithm-implementation variants, as follows. (i) Spectral method using the MATLAB `eigs` function, (ii) MATLAB/Gurobi on the permutahedron-based convex formulation, (iii) MATLAB/Gurobi on the Birkhoff-based convex formulation with $p = 1$ (that is, formulation (7) with $Y = \pi_I$), and (iv) Experimental MATLAB code provided to us by the authors of [2] implementing FISTA, for solving the matrix-regularized Birkhoff-based convex formulation (7), with projection steps solved using block coordinate ascent on the dual problem. This is the current state-of-the-art algorithm for large instances of the Birkhoff-based convex formulation; we refer to it as RQPS (for "Regularized QP for Seriation"). We report run time data using wall clock time reported by Gurobi, and MATLAB timings for RQPS, excluding all preprocessing time. We used the bitonic sorting network by Batcher [13] for experiments with the permutahedron-based formulation.

**Linear Markov Chain.** The Markov chain reordering problem [2] involves recovering the ordering of a simple Markov chain with Gaussian noise from disordered samples. The Markov chain consists of random variables $X_1, X_2, \ldots, X_n$ such that $X_i = bX_{i-1} + \epsilon_i$, where $b$ is a positive constant and $\epsilon_i \sim N(0, \sigma^2)$. A sample covariance matrix taken over multiple independent samples of the Markov chain with permuted labels is used as the similarity matrix in the 2-SUM problem. We use this problem for two different comparisons. First, we compare the solution quality and running time of RQPS algorithm of [2] with the Gurobi interior-point solver on the regularized permutahedron-based convex formulation, to demonstrate the performance of the formulation and algorithm introduced in this paper compared with the prior state of the art. Second, we apply Gurobi to both the permutahedron-based and Birkhoff-based formulations with $p = 1$, with the goal of discovering which formulation is more efficient in practice.

For both sets of experiments, we fixed $b = 0.999$ and $\sigma = 0.5$ and generate 50 chains to form a sample covariance matrix. We chose $n \in \{500, 2000, 5000\}$ to see how algorithm performance scales with $n$. For each $n$, we perform 10 independent runs, each based on a different set of samples of the Markov chain (and hence a different sample covariance matrix). We added $n$ ordering constraints for each run. Each ordering constraint is of the form $x_i + \pi(j) - \pi(i) \leq x_j$, where $\pi$ is the (known) permutation that recovers the original matrix, and $i, j \in [n]$ is a pair randomly chosen but satisfying $\pi(j) - \pi(i) > 0$. We used a regularization parameter of $\mu = 0.9\lambda_2(L_A)$ on all formulations.

**RQPS and the Permutahedron-Based Formulation.** We compare the RQPS code for the matrix-regularized Birkhoff-based convex formulation (7) to the regularized permutahedron-based convex formulation, solved with Gurobi. We fixed a time limit for each value of $n$, and ran the RQPS algorithm until the limit was reached. At fixed time intervals, we query the current solution and sample permutations from that point.

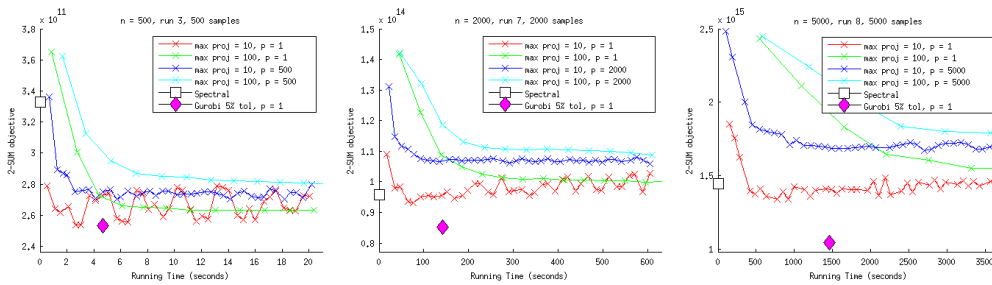

Figure 3: Plot of 2-SUM objective over time (in seconds) for $n \in \{500, 2000, 5000\}$. We choose the run (out of ten) that shows the best results for RQPS relative to the interior-point algorithm for the regularized permutahedron-based formulation. We test four different variants of RQPS. The curves represent performance of the RQPS code for varying values of $p$ (1 for red/green and $n$ for blue/cyan) and the cap on the maximum number of iterations for the projection step (10 for red/blue and 100 for green/cyan). The white square represents the spectral solution, and the magenta diamond represents the solution returned by Gurobi for the permutahedron-based formulation. The horizontal axis in each graph is positioned at the 2-SUM objective corresponding to the permutation that recovers the original labels for the sample covariance matrix.

For RQPS, with a cap of 10 iterations within each projection step, the objective tends to descend rapidly to a certain level, then fluctuates around that level (or gets slightly worse) for the rest of the running time. For a limit of 100 iterations, there is less fluctuation in 2-SUM value, but it takes some time to produce a solution as good as the previous case. In contrast to experience reported in [2], values of $p$ greater than 1 do not seem to help; our runs for $p = n$ plateaued at higher values of the 2-SUM objective than those with $p = 1$.

In most cases, the regularized permutahedron-based formulation gives a better solution value than the RQPS method, but there are occasional exceptions to this trend. For example, in the third run for $n = 500$ (the left plot in Figure 3), one variant of RQPS converges to a solution that is significantly better. Despite its very fast runtimes, the spectral method does not yield solutions of competitive quality, due to not being able to make use of the side constraint information.

**Direct Comparison of Birkhoff and Permutahedron Formulations**     For the second set of experiments, we compare the convergence rate of the objective value in the Gurobi interior-point solver applied to two equivalent formulations: the vector-regularized Birkhoff-based convex formulation (10) with $p = 1$ and the regularized permutahedron-based convex formulation (9). For each choice of input matrix and sampled ordering information, we ran the Gurobi interior-point method In Figure 4, we plot at each iteration the difference between the primal objective and $\overline{v}$.

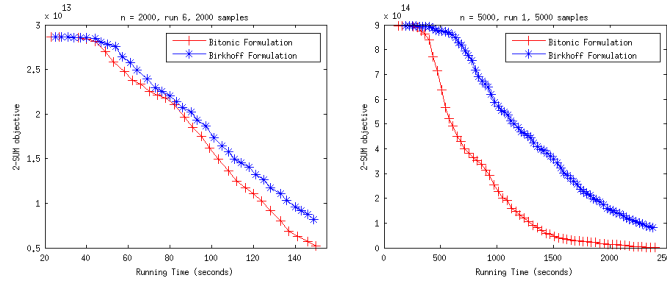

Figure 4: Plot of the difference of the 2-SUM objective from the baseline objective over time (in seconds) for $n \in \{2000, 5000\}$. The red curve represents performance of the permutahedron-based formulation; the blue curve represents the performance of the Birkhoff-based formulation. We display the best run (out of ten) for the Birkhoff-based formulation for each $n$. When $n = 2000$, the permutahedron-based formulation converges slightly faster in most cases. However, once we scale up to $n = 5000$, the permutahedron-based formulation converges significantly faster in all tests.

Our comparisons show that the permutahedron-based formulation tends to yield better solutions in faster times than Birkhoff-based formulations, regardless of the algorithm used to solve the latter. The advantage of the permutahedron-based formulation is more pronounced when $n$ is large.

# 6    Future Work and Acknowledgements

We hope that this paper spurs further interest in using sorting networks in the context of other more general classes of permutation problems, such as graph matching or ranking. A direct adaptation of this approach is inadequate, since the permutahedron does not uniquely describe a convex combination of permutations, which is how the Birkhoff polytope is used in many such problems. However, when the permutation problem has a solution in the Birkhoff polytope that is close to an actual permutation, we should expect that the loss of information when projecting this point in the Birkhoff polytope to the permutahedron to be insignificant.

We thank Okan Akalin and Taedong Kim for helpful comments and suggestions for the experiments. We thank the anonymous referees for feedback that improved the paper's presentation. We also thank the authors of [2] for sharing their experimental code, and Fajwal Fogel for helpful discussions. Lim's work on this project was supported in part by NSF Awards DMS-0914524 and DMS-1216318, and a grant from ExxonMobil. Wright's work was supported in part by NSF Award DMS-1216318, ONR Award N00014-13-1-0129, DOE Award DE-SC0002283, AFOSR Award FA9550-13-1-0138, and Subcontract 3F-30222 from Argonne National Laboratory.

## Footnotes

[1] In [2], each column of $Y$ is said to contain a perturbation of $\pi_I$, but in a response to referees of their paper, the authors say that they used sorted uniform random vectors instead in the revised version.

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
