[Reviews · NeurIPS 2014]

Submitted by Assigned_Reviewer_6

The authors derive a new convex relaxation for the noisy seriation problem (a combinatorial ordering problem, where variables must be ordered on a line such that their pairwise similarities decrease with their distance on this line).
Specifically, they use the construction in Goemans [1] based on sorting networks, in order to optimize over the convex set of permutation vectors (ie. the permutahedron) instead of the convex hull of permutation matrices (ie. the Birkhoff polytope). The new representation reduces the number of constraints from Theta(n^2) to Theta(nlog^2n) and turns out to be in practice significantly faster to solve some instances of the seriation problem.

I think this paper provides a very appealing convex relaxation to the seriation problem, since it enables to solve much larger instances (up to several thousands with a standard interior point solver, against to a few hundreds with previous relaxation in [2]).
A few points might need some clarification, especially:
- In the experiments, the authors vary p in the convex relaxation of [2], though it seems the parameter to adjust should be mu. This should not affect much the computation times, but may have an impact on the quality of the results? Besides, the convex relaxation of [2] is inexactly transcribed (there is a missing factor 1/p in front of the first quadratic term).
- More generally speaking, results of experiments on Markov Chain are a little bit hard to read, maybe replace the graphs by a table?
- In the end of the Experiments section, the authors mention they have done experiments on graphs, but they do not provide the detailed results in the appendix.
- A fast first order algorithm is presented in the appendix, but it is not compared with in the Experiments section
- There seems to be some confusion in the definition of R-matrices: line 153 it is claimed that only one permutation can reorder a pre-R matrix into a R matrix, but this result is trivially false (eg. take a constant similarity). In the same spirit, the authors recall inexactly a result from [2] line 172 (the result was shown true only for a subclass of R-matrices). However, this does not affect the rest of the paper.
- Figure 2 is not very meaningful to me

I mention to the authors a few typos: line 65 “values values”, line 308 “we experimented too …”, line 320 “it is ideal…”, line 334 “abd”, line 421 “our work would (will) spur…”

Quality

The authors provide good references to support their analysis, and some promising experimental results, in terms of computational cost. However, some of the R-matrix properties seem to be misunderstood, and the experimental results could be better presented.

Clarity

The paper is written in a clear way, though it could be polished (cf. typos).
The regularization of the convex relaxation of [2] needs some clarification (confusion between p and mu).
Moreover, more details on the sorting networks would be welcome to make the paper more self-contained.

Originality
This paper gives a novel combination of previous works to solve an ordering problem in a much faster way.

Significance

This new relaxation might be useful for researchers interested in ordering or related optimization problems. It would be interesting to see experiments on other real datasets (e.g. DNA sequencing, ranking etc.).

Summary: This paper provides a computationally cheap convex relaxation to the 2-SUM combinatorial ordering problem, using a representation that had not been used before (sorting networks) in this context. Some parts need some clarification (experiments, R-matrix, sorting network description…).

*** added after rebuttal ***
I have read the author response and happy of the answers to my comments. Thanks

Submitted by Assigned_Reviewer_11

This paper considered the sorting network polytope for the convex relaxation of permutation problems. As the permutahedron has exponential number of facets which is computationally infeasible, the permutation problems instead are relaxed to convex optimization problems with Birkhoff polytope in the literature. The Birkhoff polytope has O(n^2) constraints. Using Geoman's compact formulation of permutahedron, this paper further relax the problem with O(nlogn) constraints. The new formulation based on sorting network polytope is applied to noisy seriation problem, which can be solved by the relaxation of 2-SUM problem.
Extensive experiments were done in matlab, CVX and Gurobi, where the proposed method was compared to spectral method and Birkhoff polytope based convex formulation on different datasets. Experiment results show the superiority of the proposed method in computation time. Overall, the paper is well written. The contribution is clearly stated.

Typos: line063, two stop points. line335 "abd"->"and"
Summary: The paper uses the sorting network polytope to replace the Birkhoff polytope for the convex relaxations of permutation problems, reducing the number of constraints from O(n^2) to O(nlogn). The new formulation can achieve similar quality results in less computation time, which is supported by the experiments.

Submitted by Assigned_Reviewer_41

This paper studies the seriation problem where n objects with known characteristics must be ordered in a sequence such that a prescribed similarity measure of the characteristics decreases with the distance of the objects in the sequence. The noiseless seriation problem is formalized as the recovery of an R-matrix (whose entries are non-increasing as we move away from the diagonal horizontally or vertically) from a similarity matrix that has the structure of a pre-R-matrix (i.e., an R-matrix with permuted rows and columns). In the noisy seriation problem the similarity matrix is only approximately equal to a pre-R-matrix, and thus no exact R-matrix can be recovered.

The noiseless seriation problem can be formulated as a 2-SUM optimization problem, which minimizes a convex quadratic form over all permutations of the integer vector (1,2,...,n). As the 2-SUM problem is NP-hard, this paper suggests to solve a convex relaxation of the 2-SUM problem over the convex hull of all permutations. It is argued that such a relaxation can also offer high-quality solutions for the noisy seriation problem. This approach requires a heuristic to recover a valid permutation from the solution of the relaxed problem.

The convex hull of all permutations of (1,2,...,n) can be seen as a projection of the Birkhoff polytope of permutation matrices, which has a representation in terms of n^2 variables and constraints. This approach was adopted in reference [2]. The present paper suggests to use a more efficient sorting network representation of the convex hull of all permutations, which was discovered in [1] and only involves n*log(n) variables and constraints. While the noiseless seriation problem can be solved exactly and efficiently with a spectral method, the proposed relaxation method (as well as the relaxation method from [2]) provide good solutions for the noisy problem and when partial prior information about the sequence of the n elements is available (e.g., when some elements satisfy known precedence relations).

The paper addresses an important topic with interesting applications in machine learning. The idea to use more efficient descriptions of the permutahedron is a good one, and the numerical results clearly demonstrate the merits of this approach. On the theoretical side, the contributions seem less convincing. It appears that this paper essentially combines an idea from [2] to solve the seriation problem by seeking convex relaxations of a related 2-SUM problem and from [3] to represent the permutahedron using a sorting network. Once these two ideas are combined, everything else is more or less straightforward. This is maybe the biggest weakness of the paper.

I did not spot any mathematical errors or inaccuracies.

The paper is generally well written and well structured. I just have some minor comments about the exposition:
- l128: The sentence "Theorem 2.3 can be used to show that this construction can be used" sounds awkward.
- l169: It would be useful to remind the reader that the Laplacian matrix is always positive semidefinite because it is symmetrical and diagonally dominant.
- p4: It sounds as if the NP-hardness of the 2-SUM problem is the reason to study convex relaxations of (5). However, it is stated that (5) is efficiently and exactly solvable by the spectral method. This sounds like a contradiction. Of course, there is no contradiction, because the seriation problem is a special case of the 2-SUM problem where the input matrix is a pre-R-matrix, and this special case is tractable. It would be good to make this clearer and to motivate the formulations (7) and (9) by the desire to incorporate prior information and to account for noisy data (in which case the spectral method may no longer be applicable or display an inferior performance). It would be useful to work out these arguments more clearly. I had to read this several times until I (hopefully) understood the logic.
- l199: I missed the definition of Pi_I. Is this the permutation matrix corresponding to the trivial permutation pi_I? So it is the identity matrix?
- Figure 2(left): I am confused by the meaning of the outer circle and how it relates to the ellipsoidal level curves.
- l291: "The asymptotically sorting network" sounds awkward.
- l334: typo: "abd" -> "and".
- Table 1: The caption states that solver times are averaged over 10 runs. However, the performance measures seem also to be averaged. Please clarify.

Summary: The paper develops a convex optimization approach for solving the seriation problem. The idea to use efficient descriptions of the permutahedron is a good one, and the numerical results clearly demonstrate the merits of this approach. It appears that this paper essentially combines an idea from [2] to solve the seriation problem by seeking convex relaxations of a related 2-SUM problem and from [3] to represent the permutahedron using a sorting network.
Author Feedback
Author rebuttal: Thanks to the reviewers for your detailed comments and suggestions! We would like to note that we have a full version of this paper that fixes all typos spotted by the referees, and has a revised and expanded experiment section. In particular, we have scaled up the problem sizes for the Markov chain problem discussed on p.8. We have also added additional comparisons between the two convex formulations (Birkhoff and permutahedron) and the two kinds of regularization (see formulas (8) and (10) in the submission) that paints a more complete picture of the differences between these variants and their computational behavior. The extended version goes part way to addressing some of the concerns of the referees. We address these and other concerns below. (We will not respond here to minor points where we agree with the reviewers and we will make the appropriate corrections in the revision.)

[Assigned_Reviewer_41]

(Comment/Question):
l199:I missed the definition of Pi_I. Is this the permutation matrix corresponding to the trivial permutation pi_I? So it is the identity matrix?
(Response):
This was a typo; we meant pi_I.

(Comment/Question):
Figure 2(left):I am confused by the meaning of the outer circle and how it relates to the ellipsoidal level curves.
(Response):
The outer circle represents the points with the same l2-norm as the permutation vectors. The spectral solution is given by where the level curves meets this circle.

(Comment/Question):
Table 1:The caption states that solver times are averaged over 10 runs. However, the performance measures seem also to be averaged. Please clarify.
(Response):
We meant to say that all the measures were averaged. (In the extended version, we have dropped timings for this dataset altogether, since for most formulations the times are fast, with the one exception being due more to an infelicity in the CVX solver than to a defect in the formulation.)

[Assigned_Reviewer_6]

(Comment/Question):
In the experiments, the authors vary p in the convex relaxation of [2], though it seems the parameter to adjust should be mu. This should not affect much the computation times, but may have an impact on the quality of the results? Besides, the convex relaxation of [2] is inexactly transcribed (there is a missing factor 1/p in front of the first quadratic term).
(Response):
We agree with the referee regarding variation of p. We were constrained in the submitted version by using CVX to solve all formulations on the Munsingen data set, and CVX was unacceptably slow on the Birkhoff formulation. We have switched to using Frank-Wolfe for all formulations on Munsingen, since timings and efficiency are not of significant interest here. In the revised Munsingen experiments, we have compared a larger set of variants, in which we vary p, mu, the choice of regularization (the “matrix-based” regularizer of (8) and the “vector-based” regularizer of (10)), and the choice of formulation (Birkhoff or permutahedron). We obtain superior results for the permutahedron-based formulation, and the vector-based regularizer, so our claims for these new formulations hold up.
(Omission of 1/p is a typo.)

(Comment/Question):
More generally speaking, results of experiments on Markov Chain are a little bit hard to read, maybe replace the graphs by a table?
(Response):
In the extended version we have “torn apart” these graphs to make them less complicated and easier to interpret. We include additional graphs that illustrate the difference between the Birkhoff and permutahedron formulations, both solved with Gurobi.

(Comment/Question):
In the end of the Experiments section, the authors mention they have done experiments on graphs, but they do not provide the detailed results in the appendix.
(Response):
We were not able to obtain comprehensive on graph problems by the deadline. Subsequently, we decided not to pursue results on the graph data that was used in [22], as the case for solving the seriation problem on this graph was weak, and no “baseline” solution was available. Instead, we solved larger instances of the Markov chain problem (of order similar to the graph in [22]) and give detailed results for these instances in the extended version of the paper.

(Comment/Question):
A fast first order algorithm is presented in the appendix, but it is not compared with in the Experiments section
(Response):
The first-order algorithm in the appendix applies to the formulation in which just a single tiebreaking constraint is present (and no other ordering information), whereas side constraints are present in most of our experiments. We have included it as a matter of possible theoretical interest.

(Comment/Question):
There seems to be some confusion in the definition of R-matrices: line 153 it is claimed that only one permutation can reorder a pre-R matrix into a R matrix, but this result is trivially false (eg. take a constant similarity). In the same spirit, the authors recall inexactly a result from [2] line 172 (the result was shown true only for a subclass of R-matrices). However, this does not affect the rest of the paper.
(Response):
Thank you for catching these errors in our exposition on R-matrices. We had actually said that for any pre-R matrix there is only one corresponding R-matrix, but even this claim is false (consider any diagonal matrix with non-duplicated entries). We agree about our recollection of the result from [2] and will fix this.

(Comment/Question):
Figure 2 is not very meaningful to me
(Response):
Figure 2 tries to provide intuition about the differences between the relaxations and the spectral solution. We believe the figure is worth including and will try to provide a better explanation in the revision.